# Guidelines to Establish an Office of Student Accessibility Services in Higher Education Institutions

**Ricardo Mendoza-González** [1,*], **Sergio Luján-Mora** [2] , **Salvador Otón-Tortosa** [3] , **Mary Sánchez-Gordón** [4] , **Mario Alberto Rodríguez-Díaz** [1] **and Ricardo Emmanuel Reyes-Acosta** [1]

1. Department of Systems and Computing, Campus Aguascalientes, Tecnológico Nacional de México, Aguascalientes 20256, Mexico; mario.rd@aguascalientes.tecnm.mx (M.A.R.-D.); ricardo.ra@aguascalientes.tecnm.mx (R.E.R.-A.)
2. Department of Software and Computing Systems, University of Alicante, 03690 Alicante, Spain; sergio.lujan@ua.es
3. Computer Science Department, University of Alcala, 28871 Madrid, Spain; salvador.oton@uah.es
4. Computer Science Department, Østfold University College, 1757 Halden, Norway; mary.sanchez-gordon@hiof.no
* Correspondence: mendozagric@aguascalientes.tecnm.mx

**Abstract:** The objective of this paper is to propose a set of guidelines to establish an office of Student Accessibility Services (SAS) in Higher Education Institutions (HEIs). The proposed guidelines help to integrate disjointed knowledge to facilitate its interpretation and implementation during deployment of basic support services in favor of students with disability. These guidelines can help to mitigate complexity in providing SAS for the first time in HEIs. These guidelines cover both the design and implementation of an office of SAS and its management. Knowledge was found through a multivocal literature review (MLR), which allowed to capture not only academic approaches but also vantage points and experiences from practice. Key concepts and aspects were organized into eight components (five related to the design and implementation, and three associated with the management context). An expert appraisal method was used as a proof of concept, which complemented a previously performed preliminary implementation example. Obtained results demonstrated the pertinence of the conceptual proposal and confirmed guidelines capability for full implementation in a real-world scenario.

**Keywords:** office of student accessibility services; accessible education; virtual learning; higher education

## 1. Introduction

The Convention on the Rights of Persons with Disabilities (CRPD) highlights the right of people with disabilities to access education without discrimination and equal opportunities [1]. In general terms, accessibility in education means that a person with disabilities must be able to "*acquire the same information, engage in the same interactions, and enjoy the same services, in an equally effective and equally integrated manner, with substantially equivalent ease of use, as a person without disabilities*" [2]. In higher education institutions (HEIs), this implies, as broadly as possible, environments (including virtual), processes, access to information, objects, tools, devices, communication, goods, and services, by considering universal design principles and reasonable adjustments [3–5].

On the other hand, accessibility and inclusion in education represent key aspects for sustainability development [6]. According to the 2030 Agenda for Sustainable Development, the fourth Sustainable Development Goal (SDG) is Education. This goal is [7] (i) to "*ensure inclusive and equitable quality education and promote lifelong learning opportunities for all*" by providing access to affordable and quality technical, vocational, and tertiary education, including universities (Target 4.3); and (ii) to ensuring equal access to all levels of education and vocational training for the vulnerable, including persons with disability (Target 4.5).

Even some experts and disability advocates agree that all seventeen SDGs and objectives from CRPD are related to accessibility and disability, either directly or indirectly [6,8]. According to [9], accessibility is also one of the measures of social sustainability. However, accessibility in universities can be complicated in practice since higher education is broadly considered as the most exclusionary educational level [10]. Besides that, what is accessible to someone may not be for another person, therefore, it becomes important to analyze not only if certain alternatives are accessible or not, but also understand to whom they are accessible, under what conditions, and for which tasks [2].

In this way, student accessibility services (SAS), commonly embodied as offices or units, have emerged during the past three decades in universities intended to achieve truly accessible higher education [10]. The office of SAS is also known as the Office for Students with Disabilities, Office of Disability Services (ODS), Student Disability Services, or Office for Inclusive Education. These offices are focused on guaranteeing the full inclusion and participation of university students with disabilities considering their individuality through effective equal opportunities and non-discrimination in academic life. Offices of SAS also promote sensitization and awareness of all members of the university community [11].

In general, an office of SAS could be perceived as a way to foster and implement diverse accessible strategies, assistance, and support in favor of students with disabilities [12]. The office of SAS is closely related to virtual environments and ICT (Information and Communications Technology). In this context, accessible ICT are implemented to empower support to students and faculty with or without disabilities through virtual campus, 3D job simulators, or Open Educational Resources (OERs) [13].

These aspects suggest that the establishment of one of these offices in universities could be difficult, especially for those institutions new in the accessibility context, since many factors are involved, such as training, technology, and collaboration [10,11]. The successful implementation of accessibility services requires the involvement of the entire university community and its holistic integration with the institution. In this situation, it is evident that there is a need for standardized guidance to mitigate the special educational needs of students with disabilities in structured similar ways at different higher education institutions (HEI) [10].

In order to contribute to reducing these gaps, the proposed guidelines in this paper are intended to provide basic but reliable recommendations to establish an office of SAS in HEI, considering two specific objectives.

1. To provide guidance to design and implement an office of SAS in HEIs without these kinds of services.
2. To offer information useful to manage and operate an office of SAS in HEIs.

In this way, the proposed guidelines integrate best practices identified in literature considering main features of current offices of SAS in universities mainly from countries like Spain and the United States (even if no geographical criteria were defined for the study, the most of significant sources identified came from these countries). The information includes support services provided, internal organization, people involved, and quality strategies.

The content of this paper is organized as follows. Section 2 offers additional information about the office of SAS, including first attempts of support services, objectives, and ideal features. Section 3 describes how the guidelines were developed and Section 4 provides an overview of the guidelines. Section 5 summarizes the expert validation of the proposal. Finally, Section 6 offers main concluding remarks and future work.

## 2. Background

In 1948, the University of Illinois at Urbana-Champaign became the first post-secondary institution to provide a support service program enabling students with disabilities to attend, which evolved during the ensuing years to the current Disability Resources and Educational Services (DRES) at the University of Illinois [14]. Currently, DRES is ranked in the Top 10 of the best disability-friendly colleges and universities, sharing the honor with otherssuch as the Department of Disability Resources from Texas A&M, the Office of Acces-

sible Education from Stanford University, Services for Students with disability Office at the University of Michigan, and the Student Disability Services from Cornell University [15].

Similar efforts have emerged around the world; for example, in Spain, in 1989 the Universidad Autónoma de Barcelona was the first public university in offering attention services for students with disability and was followed by many others such as the Universidad de Alicante in 1997 and Universidad de Alcalá in 2005 [16]. In Mexico, some public universities were pioneers in integrating SAS, including the Universidad Autónoma de México, Universidad Veracruzana, and Universidad Autónoma de Tlaxcala, which offer grants for students with disabilities as part of their services (Becas de discapacidad para entrar en la universidad en México, Universia MX, https://www.universia.net/mx/actualidad/becas-y-ayudas/becas-de-discapacidad-para-entrar-en-la-universidad-en-mexico.html accessed on 20 December 2021).

In this context, a broad study to understand the degree of inclusion in Spanish universities revealed some specific objectives to be covered by offices of SAS in HEIs [11]:

- Facilitate access to university studies for pre-university students with disabilities;
- Provide information, training, and support to the university community in the effective application of inclusion policies and regulations for people with disabilities;
- Offer resources and academic advice to students with special educational needs derived from their disability;
- Guarantee accessibility to university spaces including virtual information, services, and learning;
- Collaborate with different institutional levels in the university, as well as with external organizations and entities to improve the effectiveness of the services;
- Collaborate and contribute to the labor insertion of students with disabilities and observe contract regulations in favor of people with disabilities during students' recruitment;
- Promote awareness in the university environment regarding people with disabilities;
- Update knowledge through training programs for those professionals in the public and private sectors who care for people with disabilities.

Similarly, international regulations foster the relevance of student accessibility services by asking universities to attend students' needs through reasonable adjustments, personal assistance, accessibility awareness and sensitization, faculty training, labor insertion, technical and economic support, accessible environments, and even research on accessibility [10,16]. Fundación Universia highlighted the following ideal features of SAS [17]:

1. The service operates directly as a unit, office, or internal/specific area in the university. Although this is the most common form of operation, it can also be operated through a foundation, or as a shared task between two or more offices or internal areas in the university.
2. The office organically depends on a vice-rectory (most common organic structure), management, or another administrative area, or a foundation.
3. The office usually integrates multidisciplinary (psychologists, pedagogues, physiotherapists, sign language interpreters, among others) and inclusive (men, women, personnel with disabilities) working groups.
4. The office works together with its peers in other universities, other care units in the university itself (for example, psychological care unit), and public or private organizations that work with disabilities.
5. The office encourages the voluntary participation of students, recognizing with academic credits their collaboration in support services for people with disabilities at the university.
6. The office has a continuous improvement process or an internal evaluation system, which could be owned or based on a quality assurance standard (for example, UNE-EN ISO 9001: 2015 standard).

### 3. Materials and Methods

The MLR is a kind of systematic literature review (SLR) that considers both scientific and grey literature (materials such as unpublished studies or doctoral dissertations, conference proceedings, book chapters, government, and agency reports, as well as blogs posts, white papers, and presentation videos) enriching data with points of view, insights, and experiences into the "state of the practice" [18,19]. MLR was quite relevant for the objectives of the proposed set of guidelines since each recommendation offered by the proposal would be based on successful insights from practice, but also on scientific perspectives. The study was performed considering the process explained in [20], which provides detailed guidance for its adequate implementation and integrates strengths from previous methodologies, including those reported in [18,19]. Figure 1 summarizes the MLR process implemented for this study, including the number of sources selected in every stage of the process.

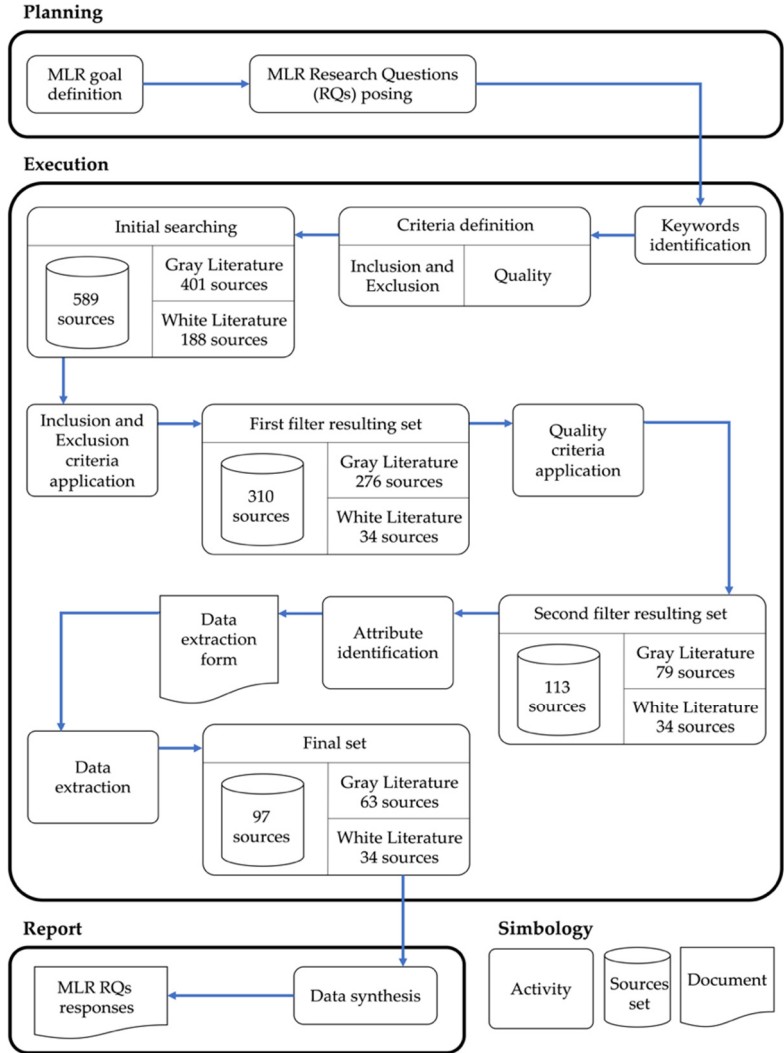

**Figure 1.** Multivocal literature review process.

### 3.1. Planning Stage

This stage was integrated by two activities, the definition of the MLR goal, and the research questions (RQs) posing as follows:

1.  Definition of MLR goal: To identify the main features (including offered services, internal organization, people involved, and quality strategies) of current SAS in universities around the world.

2.  MLR RQs posing: Derived from the MLR goal, the following four RQs were posed to delimit the study.

    RQ1.  What kind of support services are offered by the current offices of SAS in HEIs?

    RQ2.  How current offices of SAS are integrated into the organizational structures of HEIs?

    RQ3.  What are the main features (profiles and organization) of working teams in current offices of SAS in HEIs?

    RQ4.  What kind of strategies are implemented to ensure service quality in offices of SAS in HEIs?

*3.2. Execution Stage*

This stage started with keywords identification and the definition of criteria for inclusion/exclusion of sources and the quality criteria. The next step was the initial search of sources. Then, collected sources were analyzed by using defined criteria to obtain a refined set of sources. A set of attributes were identified and integrated into the extraction form, which helped to confirm the final set of sources ready for data extraction. A more detailed explanation is presented below:

-   Keywords identification: Exploratory search strings revealed some terms useful to refine searching based on the MLR goal and RQs posed. After five iterations, the following seventeen keywords were identified: accessibility service, accessibility unit, office of student accessibility services, accessibility center, counseling service, support service, technological accessibility, accessible technology, college, university, institute of technology, accessible HEI, students with disability, disability, accessibility, and inclusion. For this study, keywords were translated to Spanish to also perform searching in this language, which helped to expand the search space and reduce the risk of missing relevant sources from Spanish universities and organizations.

-   Criteria definition. The following three criteria were defined:

    1.  Inclusion criteria: Gray Literature considered for this study include unpublished studies or dissertations (doctoral, master, or undergraduate), conference proceedings, book chapters, government and agency reports, blogs posts, white papers, HEIs information on SAS, and other documents reported between January 2016 and May 2020. White Literature included research papers, formal books, conference papers, science magazine papers, and other formal scientific publications between January 2016 and May 2020. All selected sources treated the accessibility topic in the specific context of HEIs, or in a wide scope of implementation.

    2.  Exclusion criteria: For this study were excluded those sources focused on educational contexts other than HEIs. For example, those sources related to K-12 (a term used in education in the United States, Canada, and other countries to refer to pre-university educational degrees, from kindergarten to 12th grade).

    3.  Quality criteria: Quality assessment of sources was conducted through an adapted version of quality criteria suggested in [20]. In this case, specific criteria: the authority of the producer, methodology, objectivity, date, novelty, and outlet type– were considered for Gray Literature. These criteria were scored according to an accomplishment value associated with three possible responses: 1, completely; 0.5, partially; and 0, no response. The source of the gray literature could obtain a maximum score of 6 with a nominal value of 1 (total score divided by the number of assessment criteria) [20].

-   Initial searching: An initial set of 589 sources was integrated by using identified keywords as part of search strings in different search engines. Google's search engine was used to search for gray literature, identifying 401 potential sources. Adapted search strings were used to seek white literature sources in digital libraries, including ERIC, Redalyc, Whiley Online Library, Scielo, and Google Scholar, which were selected

considering the context and the main topic of the study, leading to the identification of 188 potential sources.

- Inclusion and exclusion criteria application: The initial set of 589 sources was filtered through the inclusion and exclusion criteria resulting in a set of 310 sources (276, gray literature, and 34, white literature). These criteria were implemented through discussion groups conducted by four authors of the study.
- Quality criteria application: The first filtered set of sources (310 sources) was simultaneously assessed by two authors of this study using the six quality criteria based on [20]. The resulting set represented the second filtered set and included those sources rated with a nominal value $\geq 0.67$. This set integrated 113 sources (79, gray literature, and 34, white literature). The reliability of this assessment was determined by applying the Krippendorff alpha, obtaining an ordinal alpha of 0.838, which was in the range of acceptable reliability values [21].
- Attribute identification: According to [22], and considering the MLR goal and the four RQs, eight transversal concepts and their related specific attributes were identified from the second filtered set of sources (113 sources). This centric concept strategy helped to classify and organize a framework for the literature review, which was then complemented by a concept matrix to analyze each source. The concepts and attributes identified for the literature review are presented below:
  - (a) The main contribution, nine attributes: description, guidance, process, methodology, tool, norm, law, state of the art, other.
  - (b) Treatment of accessibility/disability as a topic, three attributes. Specific topic, transversal topic, not mentioned in the content.
  - (c) Kind of resource, six attributes. Solving proposal as design or development, model implementation, model assessment, experience, opinion/insight, other.
  - (d) Country.
  - (e) RQ1, kind of support services offered by the office of SAS, six attributes: curricular adjustments, reasonable adjustments, student mobility, personalized strategies, fostering employability, and others.
  - (f) RQ2, integration of offices of SAS into HEI organization, four attributes: internally, externally, blended, other.
  - (g) RQ3, features of working teams in offices of SAS in HEIs, four attributes: multidisciplinary teams, with external collaboration, volunteering, and others.
  - (h) RQ4, quality assurance strategies, three attributes: continuous improvement process, increasing academic rigor, other.

  These aspects and attributes were structured into a data extraction form which was complemented by demographic information from each source, including link, kind of source, title, publication/elaboration year, authoring, and authors affiliation kind (academy, industry, collaboration, other).

- Extracting data: Data extracting form allows to map attributes and sources' content to conform a final set of 97 sources (63, gray literature, and 34, white literature). The final set of literature structured into the data extracting form is available online (Final set of literature structured into the data extracting form is available at URL https://figshare.com/s/e80b39f378f68044ed86 accessed on 11 February 2022).

### 3.3. Report Stage

The report stage of the study included the data synthesis and the responses for RQs. In data synthesis, the final set of 97 sources (63, gray literature, and 34, white literature) was organized according to its relevance considering the accomplishment of attributes in data extraction form. Organized data revealed the following quick responses for MLR RQs:

RQ1. What kind of support services are offered by the current offices of SAS in HEIs? Study suggested that reasonable adjustments (such as accessible technology, tutoring, faculty training on accessible learning, workspaces adaptation, and accompaniment) were the

most popular support service offered by HEIs in favor of students with disabilities, these kinds services were detected in 84 sources while curricular adjustments were mentioned in 70 sources, considering exams and evaluations adaptation, curricula adaptation, and classroom activities. Personalized services for students with disability were founded in 32 sources. These services were closely related to curricular and reasonable adjustments complemented by ad-hoc features, for example, specific accessible classroom furniture; places for service dogs or assistance/companion animals; volunteers for note-taking and class recording. Strategies for student mobility supported by grants from government and/or industry were mentioned in 10 sources. Similarly, alternatives to foster employability were detected in 10 sources. Finally, 5 sources complemented some of the previous services with others, including accessible virtual learning, own accessible software, and accessible institutional websites.

RQ2. How current SAS offices are integrated into the organizational structures of HEIs? Most sources (61) suggested that SAS are managed and operated internally by HEIs themselves. On the other hand, 14 sources reported a blended organization by considering a good practice to include external consultants particularly to conform an accessible virtual campus.

RQ3. What are the main features (profiles and organization) of working teams in current offices of SAS in HEIs? Most of HEIs (76 sources) reported multidisciplinary working teams to operate its office of SAS. Profiles detected include psychologists, pedagogues, experts in tutoring, physiologists, medics, experts in disability, and experts in accessible technology. External collaboration was identified in 21 sources, including collaboration with other HEIs, foundations, organizations, and institutions. Finally, 15 sources suggested volunteering strategies for accompaniment and taking notes.

RQ4. What kind of strategies are implemented to ensure service quality in offices of SAS in HEIs? Continuous improvement processes were the main strategy reported (64 sources), for example, using a tutoring program to preserve the quality of services. In the same way, 45 sources suggested tutoring, monitoring, and advising to observe academic rigor and improve students' learning processes. Other similar strategies, mostly in the form of blended alternatives, were identified in 39 sources.

## 4. Proposal Description

Obtained responses for MLR RQs were synthetized into eight components which integrated guidance, mainly from practice, on adequate implementation of strategies distinguished by their effectiveness (alternatives effective in mitigating the disadvantages faced by a student with a disability) and practicability (insights that foster consistency with the available resources, institutional accessibility goals, internal guidelines, the time required to implement the adjustments, and the lack of impact on other students and/or faculty) [23]. Every component was associated with the two specific objectives of the proposed guidelines which are described in the following sections. Figure 2 summarizes this structure.

### 4.1. Recommendations to Design and Implement an Office of SAS in HEIs

One of the initial aspects for the design and implementation of an office of SAS consists of establishing a catalog of support services aligned with institutional objectives on accessibility and inclusion. However, there is a variety of support services that could be offered through the office of SAS, and their integration into a specific catalog would largely depend on the availability of resources. In this context, the following set of recommendations is intended to help HEIs through the design and implementation of an office of SAS by offering insights to establish a catalog of support services.

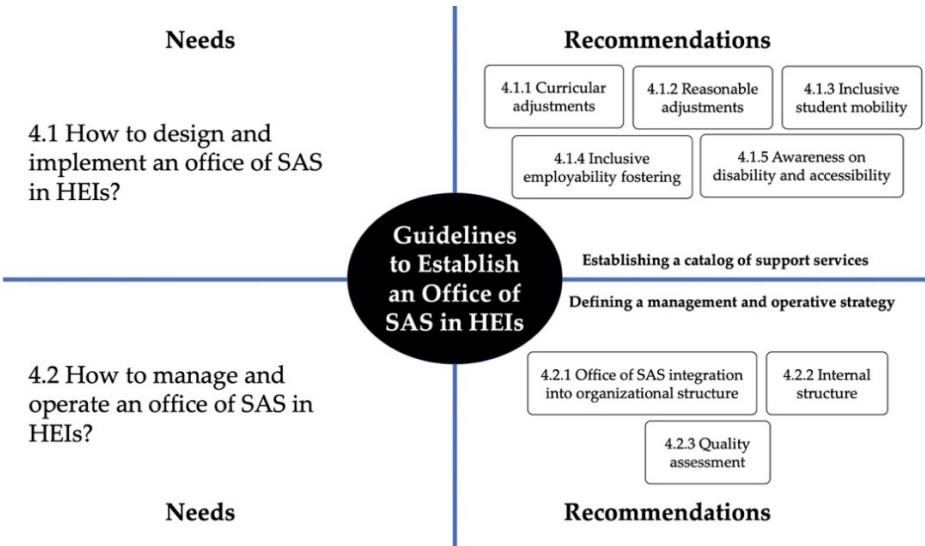

**Figure 2.** Summary of guidelines description structure.

4.1.1. Curricular Adjustments

This service involves any teaching action that facilitates the adjustment of subjects to the student who needs it. In this case, the adaptation in the methodological elements, activities, evaluation, or even some contents, could be considered an accessibility and leveling mechanism [24]. Curricular adjustments are commonly classified into two categories:

1. Curricula access adjustments, which involve adaptations and provisions of resources to lowering access barriers in both curricula and architectural spaces, for example, implement augmentative and alternative communication systems, technologies for personal mobility, most suitable places in classrooms and labs, or adaptations to furniture and spaces [25–28].
2. Curricular adjustments themselves which are subdivided into: (a) significant adaptations, consisted in modifying or replacing subjects' contents, assessment criteria and procedures, and objectives; and (b) not significant adaptations, those changes focused on accessibility consisted in adjustments to learning activities, methodologies, and to offer a variety of assessment alternatives [27–29].

It would be more convenient in higher education to maintain curricular adjustments in the context of not significant adaptations to avoid the risk of compromise professional competencies of students; nonetheless, it is important not to rule out possible significant adaptations always observing the original quality parameters of curricula and institutional regulations [24–26].

Curricular adjustments could represent a key element in ensuring inclusive education since they foster faculty awareness on specific barriers faced by students with disability and represent a means to full integration of students into the university environment [25,26,30]. Some general steps to establish curricular adaptations include [24,29]:

1. A request from a student with disability.
2. A psycho-pedagogical evaluation of the student with disabilities for information gathering, additionally, a certificate of disability may be included or required. This evaluation could be done by psychologists or professionals with a similar profile, either employees of the institution or external to the institution hired on a one-off basis.
3. Preparation of the adaptation proposal by faculty and with the support of the Office of SAS.
4. Presentation and explanation of adaptation proposal to the student with disability for its approval. It is crucial to involve students during material presentation to obtain feedback and analyze changes if needed. For example, in Universidad de Alicante (Spain), the student and faculty sign a learning contract detailing the accommodations.



The adjustments are previously established according to the type of disability (a study was carried out and regulations were drawn up with these adaptations), but they can be varied according to the specific needs of each student. These variations are introduced by the psychologist or educational psychologist of the "Student Support Center" after one or several interviews with the student and after reviewing the documentation provided (for example, medical reports).

5. Implementation of approved accommodations.
6. To offer support and follow-up service for the student.

### 4.1.2. Reasonable Adjustments

Reasonable adjustments represent those necessary and appropriate modifications and adaptations of the physical, social, and attitudinal environment to the specific needs of people with disabilities that do not impose a disproportionate or undue burden when required in a particular case in an effective and practical way to facilitate accessibility and participation of people with disability, and to guarantee the enjoyment or exercise of all rights on equal terms with others [4]. Mapping this concept to the scope the proposed guidelines, it was possible to infer that each curricular adjustment integrates a set of reasonable adjustments which could require both the participation of faculty and the involvement of other members of the university community.

The premise of reasonable adjustments is to achieve an inclusive curriculum to ensure equality of conditions for all students implementing individual adaptations considering specific requirements of every student with disability, always looking for a balance between effectiveness and practicability [23]. This goal demands effective communication and synergy among faculty, the office of SAS, and other support services at HEI, and even the collaboration with external organizations, foundations, or institutions experts on accessibility topics (for example, Fundación ONCE http://fundaciononce.es/es accessed on 20 January 2022, in Spain, which is aimed to carry out support programs for people with disabilities and to promote the global accessibility by favoring the creation of environments, products, and services for all) [23,29,31].

Ideally, the curricula should be flexible from their conception; however, this rarely happens, and adaptations are often made on the fly [31]. A strategy to mitigate this situation consists of knowing the requirements of students with disabilities before the start of the course, involving students in the adaptation process, and allowing faculty to establish, as far as possible, effective adjustments [29,31].

There is a plenty variety of reasonable adjustments, but the study presented in [29] summarizes some of the most popular and effective in practice:

1. Approaching the faculty and students with disability through tutoring programs focused on fostering empathy and communication improvement.
2. To offer different assessment alternatives for students with disability (for example, giving more time to solve exams, changing the exams' format, allowing the use of computers to solve exams, allowing a short recess during the exam, and taking the exam in a quiet place), maintaining academic standards.
3. To organize physical placement and adapt infrastructure and/or furniture in classrooms by reserving seats in the first row of the classroom for students with disability, leaving enough space between seats and aisles, and conducting class from strategic places (for example, in front of students with disabilities) in order to facilitate communication.
4. Adjust teaching methodology by making accessible all the subject's information and course materials (for example, adding subtitles to videos or audio files), allowing class recording, and providing (ideally in advance) all course materials and information in digital format.
5. Implement strategies to better explain the subject's content, for example, reorganizing topics or modifying original learning activities.

6. Implement additional strategies such as the role of collaborator student, where some advanced student voluntarily helps students with disability by note-taking, using available technological resources, and/or fostering a working team.

Reasonable adjustments represent a priority response to attend students with disability ensuring access to inclusive education through personalized and effective support measures [1,32]. On the other hand, reasonable adjustments not only promote equal treatment for students with disabilities, but also benefit students without disabilities, especially those adjustments in the classroom context, for example, to provide course materials in digital format (in advance) to facilitate the understanding of key concepts [29,31].

Basic implementation of reasonable adjustments usually involves the following steps:

1. Request for adjustments by a student with a disability before the start of the course, and ideally through the office of SAS [23,29,31].
2. Properly inform the faculty about the specific situations of students with disability (before the start of the course) and the support available from the office of SAS [23,31,32].
3. Select the reasonable adjustments to implement in favor of the student with disability, analyzing strategies to maintain academic rigor [23,29–32].
4. Determine the balance between effectiveness and practicability [30–32].

### 4.1.3. Inclusive Student Mobility

This support service involves academic and social contexts and consists of generating the appropriate conditions for the secure displacement of students with limitations, disabilities, or few opportunities, for a specific period, to another university (national or foreign) to learn, work, or participate in volunteering; and return to the university of origin to complete their studies [33,34]. Inclusive academic mobility enhances opportunities for students to acquire knowledge, skills (for example, increased proficiency in foreign languages), and experiences that help to reinforce personal confidence and development that, for one reason or another, cannot be appropriated in the basic study place [34].

Inclusive academic mobility derives from mobility procedures already available at universities; however, the following aspects could help during its implementation:

1. Manage and encourage measures, regulations, and goals that complement traditional mobility strategies by ensuring the social inclusion of students with disabilities [34–36].
2. Organize information dissemination campaigns aimed to foster the participation of students with disability in the inclusive mobility program [34].
3. Integrate into the mobility program some strategies to invite and welcome students with disability from other HEIs [34,35].
4. Implement strategies to ensure the portability of those support services that foreign students with disabilities already have, or to offer similar services from host HEI [34–36].
5. Clearly inform to host HEI about the specific needs from student with disability interested in mobility (the information should be contained in the mobility application form) [34–36].
6. Hold virtual meetings (between the interested student with disabilities, and representatives of the host and home institutions) prior to the mobility, to provide information about the host university environment, reasonable accommodations available, the city to be visited (if appliable), place of accommodation, medical assistance, accessible means of transport, personal assistance, support for daily needs (for example, cooking), recreational and/or socializing places, among other aspects related to the stay in the host place [34,36].
7. Establishment of communication channels both at the host university and at the university of origin to provide support and/or assistance for students with a disability during the mobility period [34–36].
8. Encourage effective communication and collaboration through formal agreements and procedures –among HEIs, government bodies, foundations, associations, and/or programs to generate effective strategies for truly inclusive mobility [34,35].

Additionally, for aboard mobility, it is important to provide information and even training in advance to mitigate possible cultural and linguistic impacts for students with disabilities as well as help in getting acquainted with communication procedures and pedagogical processes at the host university [36,37]. In the same way, it is very convenient to enable accessible and inclusive social, sports, and recreational spaces with cultural awareness, where a wide range of students is welcomed and their involvement in activities related to their interests, needs, and backgrounds is encouraged [34,36,37].

### 4.1.4. Inclusive Employability Fostering

Typical barriers faced by students with disabilities (for example, structural barriers and insufficient planning) frequently prevent them from achieving one of the most basic goals after obtaining a university degree, obtaining a paid and secure employment, according to their professional training [38,39]. However, it is possible to mitigate this situation through practical work experience and exposure to working environments during professional studies, since these activities help students in developing professional, technical, and transversal skills (some of the more valuable skills for companies include initiative and proactivity, motivation and enthusiasm, ability to solve problems, and teamwork [36]) that are critical for a successful transition from college to employment; and frequently, to gain confidence and security in their skills and knowledge, not only to be hired but to prosper in the positions they will occupy [38–40].

While this can be seen in all students, it has a greater impact on students with disabilities [38,39]. Some basic recommendations to foster employability for students with disability (and for all students in general) include:

1. Strengthen sensitization, communication, and collaboration with companies, other HEIs, and all those involved in the job placement or internship process to establish strategies that facilitate inclusion of students and graduates with disabilities into job offers and work practices [38,39].
2. Promote job orientation programs and services, and projects to improve employability available at the university, emphasizing support for students with disabilities [38–40].
3. Foster the virtual model for practice and job offers to mitigate access barriers, such as physical, mobility, health and medical assistance, among others [41].
4. Register complete profiles of students with disability in university integrating type of disability, skills and knowledge, professional and personal interests, specific needs and requirements, and if possible psychological information, to find the best job placement or internship options [38–41].
5. Reinforce learning actions through tutoring, training, mentoring, coaching, and/or orientation programs to foster the development of transversal skills for employment, general abilities, entrepreneurship, and self-employability [38,39,42,43].
6. Encourage students with disabilities to participate in academic competitions to demonstrate their talent and creativity in different technical and vocational skills. For example, authors of [44] mentioned the Abilympics (Abilympics, URL: https://abilympics.org.au/ accessed on 21 January 2022) competition.

### 4.1.5. Awareness on Disability and Accessibility

All university communities should be aware of accessibility barriers faced by students with disability to promote effective and proactive work and coexistence [45]. In this way, training and consulting on accessible learning would help faculty, university staff, and even students to know the needs of students with disability, best practices related to relevant legislation, and other recommendations to support students with disabilities [45–47]. For this, the following aspects are suggested:

1. Establish actions aimed at providing extensive training that explains the learning difficulties faced by students with disabilities, the existence of different learning styles, and possible accommodations that can be arranged for students with disability (for example, simple actions such as allowing notes to be taken in class through an

electronic device, or offering alternatives for video presentations, could help students with disability to improve their performance in a certain subject), since changing the perception and thoughts of faculty and university staff about disability may be essential to increase the quality of education and services offered to students with disabilities [46–48].

2.  Instruct faculty and university staff to promptly inform students (including new students) about those accessible and inclusive support services available at university and accommodations available in the particular course, not to wait for students' requirements [49].

3.  Consider training on accessible learning as mandatory and incorporate it even in the processes of hiring new faculty and staff [47,49]

4.  Offer both online and in-person training to facilitate participation [49].

5.  Consider the following topics as basic for training [50–55]:

    a.  Legislation on disability/accessibility.
    b.  Support services and specialized staff are available at the university.
    c.  Practical knowledge for the implementation of strategies in compliance with laws and policies
    d.  Institutional policies on accessibility and inclusion.
    e.  Types of disabilities (including those not obvious, such as impaired vision, distraction, difficulty to remember, learning disabilities) and their specific educational needs.
    f.  Knowledge and implementation of concepts of universal design.
    g.  Culture and etiquette in terms of inclusive communication.
    h.  Rights, capabilities, and support for university students with psychiatric disabilities.
    i.  Values in the accessible and inclusive workplace.

Other actions, such as organize events to promote social interaction with students with disabilities, would contribute to making them feel more welcome and integrated into the university community and promote understanding of disability in an environment of equity in all university activities [49]. Awareness actions could be started (i) by knowing the perceptions of students with disability on how their needs have been met by the university; and (ii) by identifying areas for improvement, which could be addressed in the training program [51,54].

### 4.2. Recommendations to Manage and Operate an Office of SAS in HEIs

After determining the catalog of support services to be offered by the office of SAS, it is very important to establish how these actions will be managed and operated in favor of students with disabilities. The management of this office should focus on solving those details that guarantee the effectiveness of the offered services, in compliance to institutional regulations and policies, and considering available resources. The next three components integrate suggestions intended to provide a starting point to set up a strategy for managing and/or enhancing the office of SAS in HEIs.

#### 4.2.1. Office of SAS Integration into Organizational Structure

The Office of SAS should be strategically incorporated into the organizational structure of HEIs in such a way that allows it to have presence in the entire university environment (all faculties, research centers, and university dependencies) through advice, awareness, and attention in the mitigation/elimination of physical, social, and cultural barriers faced by students with disabilities or in a vulnerable situation [56,57]. In this way, the office of SAS would focus on actively participating in favor of students in said condition through actions that guarantee their rights to an equal education and their total inclusion in the university environment [58,59]. The following suggestions could be useful in achieving mentioned purposes:

1.  The Office of SAS in HEIs usually derives from a Vice Chancellor (VC), particularly, VC for academic or student affairs [16,17].

2.  HEIs tend to locate offices of SAS on their websites as a transversal service, an orientation service, as an element of the section "access to university", or as the office itself [17].
3.  The office of SAS should be authorized to conform a multidisciplinary working team which could involve the faculty (as collaborators and/or tutors), students with or without disabilities (as volunteers and/or advisors), experts on accessibility and inclusion (as collaborators and/or advisors), university staff, and even external foundations and organizations (as collaborators, advisors, and/or volunteers) [16,17,60].

### 4.2.2. Internal Structure

Normally, the barriers faced by students with disabilities can be categorized into the following aspects: infrastructure, teaching-learning process, and institutional management through the university action areas (teaching, research, extension services, and linkage) [61]. In this context, it is very convenient to establish a strategic plan developed and implemented through work subcommittees with specific individual tasks, but with a unified objective institutionally recognized and appropriated, to attend situations in a formal way through comprehensive and systematic solutions [61–65] (See also Figure 3).

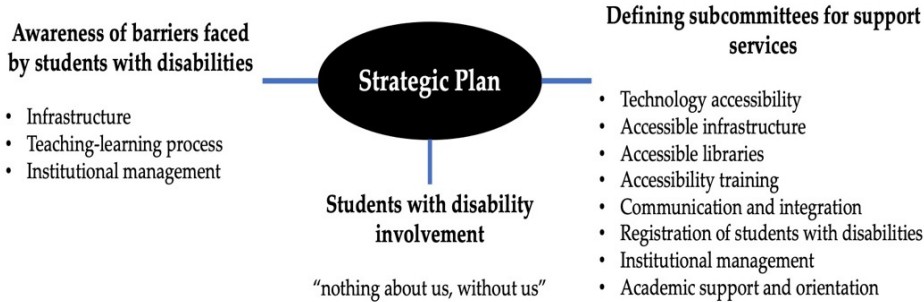

**Figure 3.** Establishing a strategic working plan for students with disabilities.

A basic set of subcommittees for the internal structure of an office of SAS could consider the following aspects, which were inferred from [62,63]:

1.  A subcommittee on technology accessibility, responsible for analyzing and promoting digital accessibility on the institutional website and the ICTs used in the university. Special attention should be paid to the accessibility of the institutional platform used as a means for digital communication between faculty and students, sending notices, grades, didactic material, learning activities, among others.
2.  A subcommittee on accessible infrastructure, focused on managing reasonable adaptations in campus spaces, as well as encouraging that each new building or space at the university campus is built with accessibility in mind.
3.  A subcommittee on accessible libraries, pursuing complete accessibility to bibliographic information through actions like the training of its staff, implementation of services such as digitization of books, access to digital libraries, and digital books complemented with screen readers.
4.  A subcommittee on accessibility training, intended to promote awareness on accessibility and disability throughout the university community to mitigate attitudinal barriers and increase understanding of the situation of students with disabilities, respect for diversity, and equal opportunities. It is also in charge of organizing interdisciplinary seminars with the participation of specialists to offer better support to students with disabilities, including technological updating, psychopedagogical attention, inclusive teaching and assessment methodologies, and generation of accessible learning materials.
5.  A subcommittee on communication and integration, focussing on establishing a direct link between the office of SAS and all university faculties. This will contribute to attend disability and accessibility situations by considering specific needs that would complement overall solutions.

6. A subcommittee for registration of students with disabilities, focussing on generating and maintaining a record of students with disabilities entering college. The registry must be systematic in order to generate reliable statistics. It may contain, for example, information about the level of ability to see, hear, move, and speak; gender; age; race; academic location (if they are already enrolled); and the socioeconomic benefit they already have (such as scholarships, and/or assistantships). In coordination with other subcommittees, it may promote and facilitate access to information in a pertinent and timely manner to make aware faculty and university staff about the condition of the students they will attend, fostering adequate educational support.

7. A subcommittee on institutional management, representing a direct link between the Office of SAS and university authorities. It is aimed to encourage a permanent institutional support for the office of SAS through a specific action protocol addressed to ensure a full accessible university environment. It is also responsible for managing financial support for students with disabilities, ensuring an inclusive context for scholarships calls (for example, there are cases of students who, due to their condition, cannot attend class regularly, so they could not cover the attendance requirement to access a specific scholarship); facilitate the completion of procedures for admission, permanence, and graduation; and design an action protocol for support service with institutional scope.

8. A subcommittee on academic support and orientation for students with disabilities, focussing on managing academic support, particularly through tutoring and volunteering. Actions include the development of a tutoring strategy for specific subjects involving faculty experts in the required topics; and the establishment of a volunteering alternative to encourage the participation of students without disability to provide assistance by taking notes, in the library, and for displacement on campus.

Additionally, it is of great value to involve students with disability in the Office of SAS' decision-making process. Their perspectives are crucial to maintaining a people-centered approach and being aligned to the popular phrase "nothing about us, without us" [60]. The following aspects (adapted from [61,62]) summarize previous suggestions to establish the internal organization of an office of SAS:

1. Identification of goals for the office of SAS.
2. Definition of the mission and vision statements.
3. Identification of the profiles for required staff.
4. Staff members recruitment.
5. Formation of subcommittees in accordance with available resources, the office of SAS goals, its mission and vision statements, and involving students with disability for decision-making from the beginning.
6. Conformation of the strategic plan among subcommittees and university authorities.

### 4.2.3. Quality Assessment

The office of SAS should clearly implement actions according to their mission and vision statements aimed at achieving their respective goals and objectives to demonstrate the value of their results as part of an institutional plan [66]. In this context, there are some resources for self-evaluating the effectiveness of offices of SAS, highlighting the following due to their current implementation in HEIs:

1. AHEAD Program Domains, Standards, and Performance Indicators (AHEAD Program Domains, Standards, and Performance Indicators, available online at URL: https://www.ahead.org/professional-resources/information-services-portal/data-collection-and-management/performance-indicators accessed on 13 February 2022) developed by the Association on Higher Education and Disability (AHEAD) represent a tool that facilitates the comparison of the practice of an office for the attention of students with disabilities, against a set of principles endorsed by the profession, which helps to determine the effectiveness of the service. The strategy involves

five domains, leadership and collaboration, consultation and information dissemination, access and equity, office administration and operations, and professional development.

2. CAS Professional Standards for Higher Education (CAS Professional Standards for Higher Education, available online at URL: https://www.cas.edu/standards accessed on 13 February 2022) were established by the Council of Advancement for Standards in Higher education (CAS). Although they are not specific to the context of attention to accessibility, they are usually used as a tool for evaluating the professional profiles of offices of SAS in HEIs.

3. External evaluators have been the most used resource over the years; however, there are no records of formal processes to contact those evaluators. They are usually contacted by recommendation [66,67].

4. Another alternative is the evaluation tool, iEvaluate OSD (Office for Students with Disabilities) [66], which provides instructions for its implementation as well as a format that facilitates its use as an evaluation tool. Such a tool is aimed at capturing the daily practices of an Office of SAS, through a questionnaire that integrates those service components considered essential by experts. The tool also suggests considering elements such as the perception of student satisfaction, the involvement of recent graduates, perception of the office of SAS by the university (faculties, departments, and other programs at the service of students), and even the participation of students without disabilities.

Similarly, the following four quality dimensions to be observed by an office of SAS are highlighted in [67]:

1. Physical facilities. Refers to the capacity of universities to provide facilities suitable for students with disabilities.

2. Access to learning. Indicates the extent to which teaching and learning, provided by university faculty, can meet the expectations of students with disabilities.

3. Communication. Measures the degree to which members of the university staff, in general, give positive feedback when providing their services to students with disabilities.

4. Empathy. Indicates the degree of sensitivity of faculty and university staff to the needs of students with disabilities.

In the same way, the Section 3: Students with Disability, from the QAA Code of Practice in Higher Education (QAA Code of Practice in Higher Education, available online URL: https://nadp-uk.org/wp-content/uploads/2015/02/2010-Code-of-practice-for-academic-qual-standards.pdf accessed on 13 February 2022) focuses on quality assurance in support services provided to students with disabilities through 24 precepts with guidance that represent the work of several leading experts in this field, and whose generality makes them susceptible to being implemented with ease in a wide range of institutions [68].

Additional aspects to consider during a quality assessment of the support services provided to students with disability in HEIs are suggested in [69], including the consciousness climate in a university campus on disability and diversity understanding; experiences, perceptions, and skills of students with disability; and the availability of support programs for the improvement of academic skills, health, and wellbeing of students with disability.

Other complementary factors that help to evaluate the quality of life of students with disabilities in the academic context are suggested in [70]: the accessibility and understanding of classes and lectures, as well as the ability to appropriate take notes and understand the course they signed up; physical accessibility to the locations where the classes and lectures are taught; and the availability of financial assistance for students with disabilities.

Support programs are frequently self-evaluated considering a set of criteria established by the office of SAS, which are commonly based on other already available criteria (as those mentioned above) [66]. In summary, these are some basic steps to implement a quality assessment strategy for the office of SAS in HEIs:

1.  Firstly, determine the set of criteria to conform to the instrument. For this, it is convenient to consider the quality criteria available in the literature, the Office of SAS goals and the mission and vision statements, the support services offered, the specific situations of the attended students with disabilities, available resources, and applicable regulations regarding attention for disability in HEIs.
2.  Establishment of a self-evaluation commission considering the participation of students with disabilities, faculty, and university staff members.
3.  Definition of the target population for instrument application, for example, only students with disability, or students with disabilities and faculty, or students with disability and students without disability, among others.
4.  Planning the self-assessment by carrying out pilot tests of the instrument, establishing dates for the application of the instrument, and establishing strategies for data analysis.
5.  Instrument application.
6.  Holding discussion meetings to identify specific improvement points and determine the strategy for results dissemination.

Even if self-evaluation in this context represents a common practice in HEIs, it is convenient to consider involving external certified experts to evaluate the effectiveness of support services offered by the Office of SAS [66–68].

## 5. Validation

The validation method by expert reviews was used to establish the conceptual validity of the proposed set of guidelines.

### 5.1. Expert Reviews

The expert review method is defined as "an informed opinion of people with experience in the subject, who are recognized by others as qualified experts in it, and that they can give information, evidence, judgments and evaluations" [71]. In this validation participated 14 experts on diverse accessibility aspects in HEIs. The accessibility aspects included accessibility assessment, research, development and/or implementation of accessible technology for education, and management of accessible strategies. The number of experts recruited is in line with other similar evaluations reported in the literature, e.g., [72,73] suggest at least five experts.

The validation was done based on the survey for assessment of conceptual models presented in [74]. This instrument consists of eight questions/statements (labeled from S1 to S8, see Figure 4) about the conceptual and methodological pertinence of the proposal considering the following aspects: scientific basis of the proposal, relevance, coverage of main aspects of the topic, logical coherence, and coincidence between the proposal and its objectives. The instrument uses a Likert scale, a simple but powerful method traditionally used in expert panel evaluations to measure the attitude or opinion of a subject, about some specific topic, with reliable results [75,76]. The Likert scale used in this study includes five options: strongly agree, agree, neutral, disagree, and strongly disagree. This instrument is publicly available online (Panel of experts survey available online at: https://forms.gle/feqwH82FmYXF9wjUA accessed on 26 January 2022). Given that all the experts were bilingual (English–Spanish), two documents were provided to them along with the survey: (i) a descriptive document of the proposal in English; and (ii) a supplemental document in Spanish which included a detailed description of the preliminary implementation of the proposal in a real scenario. Both documents are also available online (documents with proposal description used during study available online at: https://drive.google.com/drive/folders/1fpfmcIF_dYiE-GXArYXjUBRGrH4kOzRR?usp=sharing accessed on 26 January 2022).

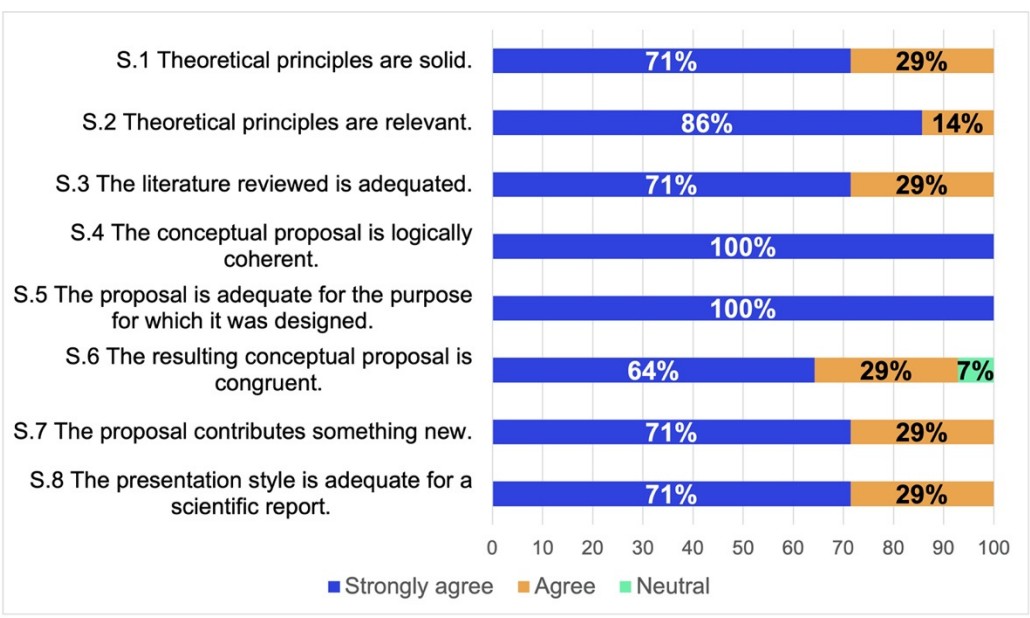

**Figure 4.** Validation results from 14 experts.

The reviewers were selected considering their background and experience on topics related to supporting services and accessibility in HEIs. Table 1 shows the demographic information of participants. It is important to mention that one participant is a person with a mobility impairment.

**Table 1.** Participants' demographic information.

| ID | Country | Gender | Experience | Category |
|---|---|---|---|---|
| Expert#1 | Canada | Male | User experience, human computer interaction | Full-time professor |
| Expert#2 | Mexico | Male | Virtual learning, accessible educational technology | Dean |
| Expert#3 | Mexico | Female | Software development, accessible virtual learning, | Part-time faculty |
| Expert#4 | Mexico | Male | Mobile apps development, usability | Full-time professor, Program Evaluator |
| Expert#5 | Mexico | Male | Electronics, artificial Intelligence, usability | Full-time professor, International Program Evaluator |
| Expert#6 | Canada | Male | Web development, artificial intelligence, educational technology | Industry, Part-time Faculty |
| Expert#7 | Mexico | Male | Virtual learning, accessible OERs | Part-time Faculty, Learning Management Systems |
| Expert#8 | Mexico | Male | User experience, human computer interaction | Full-time Professor, Program Evaluator |
| Expert#9 | Mexico | Male | Accessibility evaluation, virtual campus | Full-time Professor |
| Expert#10 | Mexico | Male | Accessibility evaluation, virtual campus | Assistant Professor, International Program Evaluator |
| Expert#11 | Spain | Male | Intelligent mobile platforms, Accessibility evaluation, virtual campus | Full-time Professor |
| Expert#12 | Spain | Female | Software engineering and human factors | Full-time Professor |
| Expert#13 | Spain | Male | Software engineering and human factors, accessible IoT | Full-time Professor |
| Expert#14 | Spain | Female | e-learning, Accessible OERs, Accessibility evaluation, virtual campus | Assistant Professor |

## 5.2. Results

Figure 4 depicts the results obtained after the expert validation.

Collected data revealed that most of the participants, 71% (10 experts), strongly agreed that this proposal was well supported by solid theoretical principles (S1). While the remaining 29% (four experts) agreed with this statement (S1). Most of the participants (12 experts, 86%) strongly agreed that relevant theoretical principles were used to develop the conceptual proposal, while two experts (14%) also agreed with the statement (S2). The

literature reviewed to develop the conceptual proposal was perceived as complete enough without presenting important omissions on the topic (S3). In this case, 10 participants (71%) were "strongly agree", and four (29%) were "agreed" with this statement.

All 14 participants were "strongly agree" that the conceptual proposal is logically coherent (S4) and adequate for the purpose for which it was designed (S5). Additionally, most of the experts (9, 64%) were "strongly agree" that the proposal is considered as congruent with the underlying research paradigm used, while four of the experts (29%) agreed, and only one participant (7%) was "neutral" with this statement (S6).

The novelty of the conceptual proposal was perceived in a positive way by the respondents (S7). In this case, 10 experts (71%) were "strongly agree" that the proposal contributes something new to the knowledge of the main topic and is not a duplication of an existing model. The remaining two experts (29%) agreed with this statement (S7).

Finally, 10 experts (71%) strongly agreed that the presentation style of the conceptual proposal is adequate for a scientific report (S8). This perception was reinforced by the remaining four participants (29%) who were "agree".

All participants were asked to optionally give an additional comment on the conceptual proposal. These additional comments also represented the bridge to further internal discussion in the project team. Nevertheless, the survey was anonymous, and it was not possible to continue the discussion with the experts because it was not possible to know which expert had formulated each comment. In total, 12 responses were received (85% of participants). The following three comments summarize the global perception from the experts:

1. *"The proposal fulfills the purpose for which it was developed. A systematic methodology was used to identify and select contributions in the literature, which adequately supports the proposal. The recommendations on the design, structure, management, and continuous improvement of a SAS Office seem to be adequate and generic for easy implementation. The example of implementation of the guidelines in the long document helped to better understand the complete proposal".* Expert#14.

2. *"The proposal is very complete, covers both the set up and running of the service, and has been tested in a real environment. Nothing to add".* Expert#12.

3. *"As one can be seen from my answers, I consider that the conceptual proposal is well supported by good theoretical principles, coherent, adequate, [the conceptual proposal] contributes to the knowledge... Maybe, the literature reviewed could be updated, although I do not know if this is possible".* Expert#10.

## 6. Concluding Remarks

Findings from an MLR allowed identifying best practices useful to establish an office of SAS in HEIs. These results were integrated into the proposed guidelines, which include essential aspects and procedures for the design, implementation, and management of an office of SAS. The proposed guidelines are intended to facilitate the integration of accessibility aspects into e-learning processes and environments in HEIs. They are focused on five aspects: (i) to visualize a catalog of basic support services in favor of students with disability; (ii) to establish orientation and training strategies for faculty and staff to foster disability awareness; (iii) to determine the integration of the office of SAS into the organizational structure; (iv) to organize the internal operational structure of the office of SAS; and (v) to establish strategies for quality assurance of the offered support services.

This conceptual proposal was validated by fourteen experts. The results showed that this proposal is supported by solid and relevant theoretical principles which highlight the importance of considering training, technology, and collaboration, to achieve a proper implementation of services that lead to truly accessible higher education. Furthermore, the validation revealed that recommendations and guidance provided in this proposal would contribute to ensuring equal opportunities for students with disability and encourage their full inclusion and participation in university life.

The validation results suggested that the proposed guidelines may contribute to mitigate the need of standardized guidance to provide educational support to students with disabilities in HEIs [10].

Additionally, the validation results helped to corroborate and support the findings of a preliminary implementation of the proposed guidelines in a real scenario (as part of the Erasmus+ project "EduTech" (EduTech—Asistencia Tecnológica a la Accesibilidad en la Educación Virtual. URL: https://edutech-project.org/ accessed on 13 February 2022). It is important to mention that these findings are not detailed in this paper; nevertheless, they were included as part of the materials for the evaluation to better explain and complement the proposal.

The findings of this study will be implemented in the four Latin-American partner universities of the "EduTech" project. This future work will allow us to analyze the full implementation of the proposed guidelines in practice. Additionally, it represents the opportunity to engage in further discussions that reveal improvement points and a better understanding of the proposal as part of the solution for the contextual problem.

**Author Contributions:** Conceptualization, R.M.-G., S.L.-M., S.O.-T. and M.S.-G.; data curation, R.M.-G., S.L.-M., S.O.-T., M.S.-G., M.A.R.-D. and R.E.R.-A.; formal analysis, R.M.-G., M.A.R.-D. and R.E.R.-A.; investigation, R.M.-G., S.L.-M., S.O.-T., M.S.-G., M.A.R.-D. and R.E.R.-A.; methodology, R.M.-G., S.L.-M., S.O.-T., M.S.-G., M.A.R.-D. and R.E.R.-A.; project administration, S.O.-T.; validation, R.M.-G., S.L.-M., S.O.-T., M.S.-G., M.A.R.-D. and R.E.R.-A.; visualization, R.M.-G., S.L.-M., S.O.-T., M.S.-G., M.A.R.-D. and R.E.R.-A.; writing—original draft, R.M.-G.; writing—review and editing, R.M.-G., S.L.-M., S.O.-T., M.S.-G., M.A.R.-D. and R.E.R.-A. All authors have read and agreed to the published version of the manuscript.

**Funding:** This research work has been co-funded by the Erasmus+ Programme of the European Union, project EduTech (609785-EPP-1-2019-1-ES-EPPKA2-CBHE-JP).

**Institutional Review Board Statement:** Not applicable.

**Informed Consent Statement:** Informed consent was obtained from all subjects in-volved in the study.

**Data Availability Statement:** The survey used in this study is available at URL: https://forms.gle/feqwH82FmYXF9wjUA accessed on 26 January 2022. Data related to the expert panel evaluation are available at the following URL: https://doi.org/10.6084/m9.figshare.19074029 accessed on 26 January 2022.

**Acknowledgments:** Authors acknowledge The European Commission and ERASMUS+ Programme for supporting this research work as part of the project EduTech (609785-EPP-1-2019-1-ES-EPPKA2-CBHE-JP). The European Commission's support for the production of this publication does not constitute an endorsement of the contents, which reflect the views only of the authors, and the Commission cannot be held responsible for any use which may be made of the information contained therein.

**Conflicts of Interest:** The authors declare no conflict of interest.

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
