# Peer review of "Guidelines to Establish an Office of Student Accessibility Services in Higher Education Institutions"

_sustainability, doi:10.3390/su14052635_

Round 1

Reviewer 1 Report

This paper originally contributes to the literature by proposing a set of guidelines to establish an office of Student Accessibility Services (SAS) in Higher Education Institutions (HEIs). This is a very important topic within the area of inclusive education and particularly for Higher Education. The propositions presented and discussed in the paper have arisen from a multivocal literature review process which is discussed in detail throughout the paper and were verified by utilizing the expert appraisal method which is also clearly explained in the paper.

Introduction and background are coherent, lay out the context of the study in a way that is understandable for the reader.  This also applies to the most parts of the paper since they are presented in a well-structured manner. Most of the cited references are up to date and appropriately cited.

Some minor revisions could be made regarding specific points (please see below).

(a) Criteria Definition – Exclusion Criteria ‘For this study were preferred those information sources whose content was somehow related to SAS in HEIs’. The exclusion criteria could be explained further, it is not clear for the reader what ‘somehow related’ means.

(b) In general, because there is lot of text and numbering within each section of the paper one recommendation would be that wherever feasible figures and/or tables could be used to make the information more accessible and better understandable to the reader.

(c) The results section could be expanded (e.g. in the case that an expert did not agree with the proposition/s did the authors engage in further discussions to understand why, where any additional steps taken? Where there any limitations by adopting this approach?)

(d) Concluding remarks. One suggestion is that this section could be supported with literature so that the relevant connections amongst the introduction, background and the empirical parts of the paper are presented on a more solid basis.

Author Response

Comment Response Location in revised document
(a) Criteria Definition – Exclusion Criteria ‘For this study were preferred those information sources whose content was somehow related to SAS in HEIs’. The exclusion criteria could be explained further, it is not clear for the reader what ‘somehow related’ means. Thank you for your comment. The exclusion criteria was rewritten and expressed in more detail. Line 217-221, page 5.
b) In general, because there is lot of text and numbering within each section of the paper one recommendation would be that wherever feasible figures and/or tables could be used to make the information more accessible and better understandable to the reader. Thank you for your comment. Two additional figures were incorporated (Figure 2, to summarize the structure of the description of the guides and facilitate reading in the document; and Figure 3, to summarize the characteristics of the strategic work plan of a SAS office); and 1 table (Table 1, with the demographic characteristics of the participants in the validation of the proposal). Line 352, see Figure 2. Line 658, see Figure 3. And Line 862.
(c) The results section could be expanded (e.g. in the case that an expert did not agree with the proposition/s did the authors engage in further discussions to understand why, where any additional steps taken? Where there any limitations by adopting this approach?) Thank you for your comment. The purpose of item 9, “Additional comments”, in the survey was clarified, including an explanation for a limitation to continue the discussion with the participants Line 978-980, page 20.

d) Concluding remarks. One suggestion is that this section could be supported with literature so that the relevant connections amongst the introduction, background and the empirical parts of the paper are presented on a more solid basis.

Thank you for your comment. The conclusions of the validation results were associated with the need for standardized guidelines (in the context of the proposal). The corresponding reference was included. Line 978-980

Reviewer 2 Report

The objective of this work is to propose a set of guidelines to establish a Student Accessibility Service (SAS) in Higher Education Institutions (IES). The research was developed through a Multivocal Literature Review (MLR) process, which allows focusing not only academic approaches, but also points of view and experiences from practice. The methodology is well planned and the results obtained contrasted the objectives and represented an interesting advance in knowledge.

Author Response

Thank you very much for your positive comments.

Reviewer 3 Report

The paper provides a thoroughly documented process of establishing an office of Student Accessibility Services (SAS) in Higher Education Institutions (HEIs). The paper is well structured. My comments are intended for clarification purposes:

p.1 "Even some experts and enthusiasts coincide ..." Who are the enthusiasts in this context?

General question regarding the literature search - were there any geographic criteria? The results indicate the study is applicable mainly in the context of Spain.

p. 5 Quality criteria - given that multiple researchers co-authored this paper, please explain who evaluated the quality -- all authors? If yes, what was the inter rater reliability?

p. 8 "A psycho-pedagogical evaluation of the student with disabilities for information gathering..." Who does this evaluation?

p. 8 "Presentation and explanation of adaptation proposal to the student with disability for its approval." Does the student have any say in this? They are presented with the material and what happens if they do not accept it or require changes?

p. 15 "Evaluations from experts" - How were the experts selected? Were experts with disabilities included in this process?

Thank you.

Author Response

Comment  Response  Location in revised document
P.1 “"Even some experts and enthusiasts coincide ..." Who are the enthusiasts in this context?” Thank you for your comment. We use a different term to better express the idea. Line 49, page 2.
General question regarding the literature search - were there any geographic criteria? The results indicate the study is applicable mainly in the context of Spain. Thank you for your comment. The reason why the mentioned countries are distinguished was clarified. Line 88-89, page 2.
p. 5 Quality criteria - given that multiple researchers co-authored this paper, please explain who evaluated the quality -- all authors? If yes, what was the inter rater reliability? Thank you for your comment. The participation of the authors in the evaluation of the quality of the sources during the application of the inclusion and exclusion criteria, and the quality criteria, was clarified. Line 246-254, page 6.
P.8 "A psycho-pedagogical evaluation of the student with disabilities for information gathering..." Who does this evaluation? Thank you for your comment. We clarified clarified the profile who could perform this type of evaluation on students. Line 398-407, page 9.
p. 8 "Presentation and explanation of adaptation proposal to the student with disability for its approval." Does the student have any say in this? They are presented with the material and what happens if they do not accept it or require changes? Thank you for your comment. We clarify this aspect and include an example of a procedure. Line398-407, page 9
p. 15 "Evaluations from experts" - How were the experts selected? Were experts with disabilities included in this process? Thank you for your comment. The aspects considered for the selection of participants were clarified. Additionally, a table (Table 1) with their corresponding demographic characteristics was included. Line 859-863, see Table 1, page 17.